# TRAP1 Chaperones the Metabolic Switch in Cancer

**DOI:** 10.3390/biom12060786

**Published:** 2022-06-04

**Authors:** Laura A. Wengert, Sarah J. Backe, Dimitra Bourboulia, Mehdi Mollapour, Mark R. Woodford

**Affiliations:** 1Department of Urology, SUNY Upstate Medical University, Syracuse, NY 13210, USA; wengertl@upstate.edu (L.A.W.); backes@upstate.edu (S.J.B.); bourmpod@upstate.edu (D.B.); mollapom@upstate.edu (M.M.); 2Upstate Cancer Center, SUNY Upstate Medical University, Syracuse, NY 13210, USA; 3Department of Biochemistry and Molecular Biology, SUNY Upstate Medical University, Syracuse, NY 13210, USA

**Keywords:** TRAP1, Hsp90, chaperone, post-translational modification, cancer, mitochondria, metabolism, Warburg effect

## Abstract

Mitochondrial function is dependent on molecular chaperones, primarily due to their necessity in the formation of respiratory complexes and clearance of misfolded proteins. Heat shock proteins (Hsps) are a subset of molecular chaperones that function in all subcellular compartments, both constitutively and in response to stress. The Hsp90 chaperone TNF-receptor-associated protein-1 (TRAP1) is primarily localized to the mitochondria and controls both cellular metabolic reprogramming and mitochondrial apoptosis. TRAP1 upregulation facilitates the growth and progression of many cancers by promoting glycolytic metabolism and antagonizing the mitochondrial permeability transition that precedes multiple cell death pathways. TRAP1 attenuation induces apoptosis in cellular models of cancer, identifying TRAP1 as a potential therapeutic target in cancer. Similar to cytosolic Hsp90 proteins, TRAP1 is also subject to post-translational modifications (PTM) that regulate its function and mediate its impact on downstream effectors, or ‘clients’. However, few effectors have been identified to date. Here, we will discuss the consequence of TRAP1 deregulation in cancer and the impact of post-translational modification on the known functions of TRAP1.

## 1. Introduction

Molecular chaperones of the heat shock protein-90 (Hsp90) family are involved in signal integration and the cellular stress response. These chaperones mediate cell signaling through the stabilization and activation of their substrate proteins, known as clients (https://www.picard.ch/downloads/Hsp90interactors.pdf, accessed 28 February 2022) [1]. The Hsp90 chaperone function is coupled to the ability to hydrolyze ATP, and chaperone activity can be precisely regulated by a heterogeneous group of proteins known as co-chaperones [2], as well as a diverse array of post-translational modifications (PTM) [3].

TNF-receptor-associated protein-1 (TRAP1) is the mitochondrial-dedicated Hsp90 family member and is localized to the mitochondrial matrix, inner mitochondrial membrane, and the intermembrane space [4,5,6]. TRAP1 was first identified through its interaction with the intracellular domain of the Type I TNF receptor [7], and early characterization of TRAP1 demonstrated ATP-binding ability and sensitivity to ATP-competitive Hsp90 inhibitors [8]. Despite this, TRAP1 was unable to form complexes with known cytosolic Hsp90 co-chaperones, nor could it promote the maturation of Hsp90 client proteins, suggesting a distinct mechanism of action for TRAP1 [8].

From this time, work has concentrated on the impact of TRAP1 on cellular processes, however identification of TRAP1 effectors and regulatory mechanisms of TRAP1 expression and activity are critical to understanding its biological function. TRAP1 has an established role as a master regulator of metabolic flux, and a large body of evidence has demonstrated that TRAP1 expression serves to suppress oxidative phosphorylation [9,10,11]. Further, TRAP1 also contributes to cell survival through complex formation with cyclophilin D (CypD), which regulates the opening of the permeability transition pore (PTP) [12]. These two known roles suggest a critical function for TRAP1 in maintaining cellular homeostasis [13]. Despite the critical importance of TRAP1 to these processes, the molecular mechanisms of TRAP1 function remain largely unresolved. Here, we will discuss recent advances in understanding the mechanisms of TRAP1 regulation, the impact of this regulation on TRAP1 function and downstream cellular processes, and the role of TRAP1 in cancer.

## 2. Structural Basis of TRAP1 Activity

Hsp90 family chaperones are characterized by their dimeric structure. Each of the two protomers are composed of an amino-terminal ATP-binding domain, followed by a middle domain, the primary interface for client interaction, and a C-terminal domain that allows constitutive dimerization of the protomers [14]. Hsp90 chaperone activity is coupled to its ability to hydrolyze ATP [15,16]. The ‘chaperone cycle’ begins with ATP binding to the ‘open’ conformation of Hsp90, followed by transient dimerization of the N-terminal domains of each protomer and ATP hydrolysis, and subsequent release of mature client proteins and regeneration of the ‘open’ Hsp90 dimer [17]. TRAP1 is broadly structurally similar to cytosolic Hsp90, with some notable exceptions, including a cleavable N-terminal mitochondrial localization signal and an N-terminal extension or ‘strap’ that stabilizes the ‘closed’ conformation of TRAP1 [18,19]. Asymmetrical post-translational modification and co-chaperone binding are important determinants of Hsp90 molecular chaperone function [18,20,21,22,23,24]. Interestingly, TRAP1 dimers are inherently asymmetric, and uniquely composed of one ‘straight’ and one ‘buckled’ protomer, with the buckled protomer demonstrating increased rates of ATP hydrolysis [25] (Figure 1). Recently, structural and cell-based studies have described a tetrameric form of TRAP1 induced in response to dysregulation of oxidative metabolism, although the impact of this TRAP1 state on its activity is as yet unknown [26]. Interestingly, whether TRAP1 ATPase activity is essential for the entire scope of its biological role also remains an open question [26].

## 3. Impact of TRAP1 on Cancer Metabolism

Controversially, TRAP1 has alternately been characterized as an oncogene and tumor suppressor, and it has been suggested that TRAP1 is essential for malignant transformation of cells but dispensable at later stages of tumor development [6,27]. Despite this controversy, much of the literature supports the idea that TRAP1 regulates metabolic transformation during tumorigenesis, TRAP1 is overexpressed in many cancers, and TRAP1 attenuation is detrimental to tumor cell survival [28,29,30,31,32,33]. It may be more appropriate to suggest that, similar to cytosolic Hsp90, many cancers may be ‘addicted’ to TRAP1 [34,35,36]. In fact, multiple pathways in which TRAP1 activity can drive tumorigenesis have been described (Figure 2) and will be reviewed in the following section.

### 3.1. Metabolic Regulation

The cellular energy currency adenosine triphosphate (ATP) is generated as a consequence of the complete oxidation of glucose to CO_2_ and H_2_O, and each molecule of glucose can maximally result in 36–38 ATP molecules [37]. Normal cells produce ATP primarily through cellular respiration, which describes a process in which glucose metabolism by glycolysis is coupled to the tricarboxylic acid cycle (TCA). Concurrent mitochondrial electron transport generates the electrochemical gradient that provides the force by which ATP is disseminated throughout the cell [38]. ATP generation is highly dysregulated in cancers, and many cancer subtypes supplement their ATP supply by upregulating cytosolic glycolysis, simultaneously generating additional ATP driven by the terminal fermentation of pyruvate to lactate [39]. This hyperactive glycolytic phenotype is known as the Warburg effect, and serves to support the accelerated growth of cancers through the increased synthesis of intermediates for anaplerotic metabolism and hypertrophy [40,41]. The phenotypic manifestations of metabolic dysregulation are variable and dependent on cell type and genotype, and many of the details and nuances of this differential regulation remain obscured.

Few specific biological roles and binding partners have been described for TRAP1, despite the broad understanding of its impact on metabolic flux. Two of the few described bona fide clients of TRAP1 however are subunits of electron transport chain (ETC) complexes, Complex II components succinate dehydrogenase subunit A/B (SDHA/B) [42,43,44,45], and Complex IV cytochrome *c* oxidase subunit 2 (COXII) [6,46,47]. Complex II/SDH is an iron–sulfur cluster-containing protein complex that functions to transfer electrons from succinate to coenzyme Q10-ubiquinone (Complex III) [48]. In agreement with the understanding of Hsp90 function, TRAP1 maintains SDH in a partially unfolded state [49], and TRAP1 inhibition releases active SDH, leading to an increase in its activity [27,44,50,51,52]. Further, SDH activity [44,53,54] and the oxygen consumption rate [6,55] are inversely correlated with TRAP1 expression, implicating TRAP1 in promoting the Warburg effect [56]. Notably, SDH also oxidizes succinate to fumarate and thus integrates the TCA cycle and the ETC, indicative of the broad influence of TRAP1 on mitochondrial metabolism [56,57,58].

Complex IV of the ETC converts molecular oxygen to water, and in doing so enacts the final step in generating the electrochemical gradient that supports ATP production by Complex V (ATP synthase) [59]. COXII is a downstream effector of TRAP1 function in the regulation of apoptosis, and TRAP1 regulates COXII expression [47] and activity [6]. As downregulation or inhibition of TRAP1 has been shown to destabilize COXII [46,50] and deletion of TRAP1 was associated with decreased COXIV subunit levels [60], it is possible that TRAP1 chaperoning of COXII/IV is mechanistically similar to SDHA/B. TRAP1 has also been shown to interact with the Complex V subunit ATPB, although little is known about this interaction [27].

Mitochondrial respiration drives the production of reactive oxygen species (ROS) and is responsible for most cellular ROS (Figure 3) [61]. In considering the role of TRAP1 in chaperoning SDH and COXII, TRAP1-mediated regulation of mitochondrial respiration suppresses ROS production [62], thereby contributing to the regulation of redox homeostasis, metabolic flux, and mitochondrial apoptosis.

### 3.2. Contribution to Tumorigenesis

Cancer-associated increases in TRAP1 expression suggest a role for TRAP1 in oncogenesis [30,63,64]. Indeed, TRAP1 deletion delayed tumor formation in a mouse model of breast cancer, providing direct evidence of the role of TRAP1 in tumor initiation [65]. Further, TRAP1-mediated SDH inhibition leads to accumulation of the oncometabolite succinate [58]. Increased succinate inhibits the activity of prolyl hydroxylases, which are responsible for the hydroxylation of the transcription factor hypoxia inducible factor (HIF1α), a prerequisite for recognition by the VHL-dependent E3-ubiquitin ligase machinery [66]. Succinate-dependent HIF1α stabilization and activation promotes a well-established glycolytic transcriptional program [67], demonstrating yet another function of TRAP1 in the regulation of cancer-associated metabolic dysregulation.

TRAP1 expression was found to be elevated in aggressive pre-neoplastic lesions in a rat model of hepatocarcinogenesis [68]. The master antioxidant transcription factor NRF2 was also activated in this model, and given the established role of TRAP1 in regulating intracellular ROS, TRAP1 likely participates in NRF2-driven ROS mitigation during tumor development [68]. NRF2 inhibition led to decreased TRAP1 levels independent of TRAP1 transcription [68], suggesting that post-translational regulation is essential for sustained TRAP1 expression in pre-cancerous and cancerous cells. Interestingly, pentose phosphate pathway (PPP) flux was found to be increased in this model, and was determined to be a consequence of elevated citrate synthase activity in aggressive pre-neoplastic lesions [68]. Citrate accumulation inhibits downstream metabolic enzymes phosphofructokinase and SDH and activates the anaplerotic PPP [69]. This increase in citrate synthase activity was alleviated following TRAP1 knockdown or inhibition, suggesting that citrate synthase may also be a TRAP1 client [68].

Cell cycle dysregulation is a well-established driver of tumorigenesis [70]. TRAP1 impacts the cell cycle through regulation of protein quality control in cooperation with the proteasome regulator TBP7 [71,72]. Loss of the TRAP1/TBP7 machinery leads to increased ubiquitination and degradation of the G2-M checkpoint proteins CDK1 and MAD2 and dysregulation of mitotic entry [72]. However, whether TBP7 is a client or perhaps even the first co-chaperone of TRAP1 remains to be seen.

Taken together, these data describe multiple mechanisms through which TRAP1 dysregulation can impact cellular metabolic flux and, potentially, tumorigenesis.

### 3.3. Evasion of Apoptosis

Mitochondrial involvement in cell death is mediated by the release of cytochrome *c* [73,74]. Sustained opening of the permeability transition pore (PTP) within the inner mitochondrial membrane (IMM) initiates a series of events that lead to cytochrome *c* release and apoptosis or necrosis. Upon PTP opening, particles under 1500 Da, such as ions (Ca^2+^, K^+^, and H^+^), water, and other solutes, flood the IMM, causing swelling and unfolding of the cristae and eventual outer mitochondrial membrane (OMM) rupture. Subsequent efflux of cytochrome *c* through the compromised OMM into the cytosol induces the caspase cascade [75,76]. This sustained PTP opening is known as the mitochondrial permeability transition (PT) [77], and it can be triggered by several mechanisms, including elevated ROS, Ca^2+^, or inorganic phosphate levels, as well as decreased pH or ATP depletion [78]. Interplay between these elements also plays a role in its regulation, as elevated ROS has been shown to decrease the amount of Ca^2+^ required to trigger the PTP [76].

TRAP1 attenuation induces opening of the PTP and release of cytochrome *c* [47], and expression of TRAP1 likely discourages the initiation of apoptosis through two distinct, but potentially overlapping mechanisms: (1) regulation of triggers that signal into the PTP, and (2) direct disruption of the physical mechanism of PTP opening. TRAP1 knockdown has been shown to lead to increased ROS accumulation under oxidative stress [79] and TRAP1 overexpression insulates cells against iron chelation-mediated ROS production [80]. These effects are likely a consequence of both direct and indirect roles of TRAP1 in minimizing ROS generation. TRAP1 is a direct regulator of oxidative phosphorylation through its chaperoning of Complexes II and IV of the ETC [6,44,46] and has an indirect role in quenching existing ROS, as TRAP1 expression is associated with increased levels of the reduced form of the antioxidant glutathione (GSH) [81]. TRAP1-dependent regulation of ROS generation also results in decreased oxidation of the phospholipid cardiolipin. This phospholipid is responsible for the binding of cytochrome *c* to the inner folds of cristae, and its oxidation results in an increase of free cytochrome *c* in the inner membrane space that can potentially escape into the cytosol [78].

Furthermore, TRAP1 has been shown to chaperone the calcium-binding protein Sorcin [82]. TRAP1 is also thought to be responsible for Sorcin translocation into the mitochondria, given that Sorcin lacks its own mitochondrial localization sequence [8,82]. Overexpression of Sorcin in neonatal cardiac myocytes has been shown to increase mitochondrial Ca^2+^ levels, while simultaneously decreasing cytochrome *c* release, indicating an increase in mitochondrial Ca^2+^ tolerance [83]. Therefore, the chaperoning of Sorcin by TRAP1 is important for desensitizing the PTP to Ca^2+^ levels. Understanding this regulation is particularly important for TRAP1, as Ca^2+^ can replace Mg^2+^ as a co-factor and induce an increased rate of TRAP1 ATP hydrolysis [84]. TRAP1 has also been shown to decrease ubiquitination of the mitochondrial contact site and cristae organizing system subunit 60 (MIC60) under conditions of extracellular acidosis [85]. MIC60 is a critical component of the protein complex MICOS, which is regarded as the master organizer of the IMM through the formation of contact sites with the outer membrane and maintenance of cristae junctions [86,87]. Thus, TRAP1 regulation of MIC60 contributes to its anti-apoptotic function through the preservation of mitochondrial integrity.

Proposals for the structure of the PTP have gone through various iterations, however the prevailing model is that the PTP is formed by coordinated activities of the adenine nucleotide translocator (ANT) and the F-ATP synthase [88,89,90]. Furthermore, cyclophilin D (CypD) is key to PTP regulation [12,91]. Though its role in this process is controversial, CypD peptidyl-prolyl isomerase activity is required, as is its binding to the mitochondrial peripheral stalk subunit of the F-ATP synthase [63,90,92]. In addition to attenuating the triggers that lead to PTP opening, TRAP1 has been shown to antagonize the opening of the PTP itself. There is a general consensus that TRAP1 accomplishes this by forming a complex with CypD, interfering with the ability of CypD to interact with the PTP [12,63,93] potentially at the peripheral stalk of F-ATP synthase [90].

Further, the mitochondrial chaperones Hsp60 and Hsp90 have been implicated in this process, as their association with CypD also prevents PTP opening; however, the architecture of this complex has yet to be characterized [12,63,93,94,95,96].

## 4. Post-Translational Regulation of TRAP1

Post-translational modification is critically important to mitochondrial function [97] and has previously been shown to regulate TRAP1, though relatively little is known about individual PTM sites (Table 1, Figure 4) [5,6,98,99]. A comprehensive study of cytosolic Hsp90 has demonstrated the importance of post-translational regulation to Hsp90 chaperone activity (reviewed in [3,100]), and in the absence of certain co-chaperone regulatory proteins, specific PTM events have been shown to functionally recapitulate their activity [101]. This phenomenon may be critically important for TRAP1 biology, as TRAP1 is thought to act without the assistance of co-chaperones [8,10].

### 4.1. Phosphorylation

PINK1 is a mitochondrially targeted serine/threonine kinase whose mutation and inactivation is linked to Parkinson’s disease [102]. PINK1 activity has previously been shown to be cytoprotective [103], and when exposed to H_2_O_2_, cells transfected with siRNA targeting PINK1 showed significant increases in cytochrome *c* release and apoptosis [5]. TRAP1 was shown to be phosphorylated by PINK1 and mediate PINK1 anti-apoptotic activity, as evidenced by the observation that TRAP1 knockdown sensitized cells to PINK1 attenuation [5,104,105]. Interestingly, TRAP1 inhibition leads to activation of PINK1, suggesting a reciprocal regulatory relationship [106].

TRAP1 has also been shown to interact with the mitochondrial serine protease HTRA2 in Parkinson’s disease [55]. Canonically, HTRA2 participates in mitochondrial and cellular quality control through inhibition of IAPs (inhibitor of apoptosis proteins) and induction of cell death, while loss of HTRA2 is associated with aberrant mitochondrial function and Parkinson’s disease (PD). Overexpression of HTRA2 led to decreased levels of TRAP1, suggesting that HTRA2 may play a role in regulating TRAP1 stability [55]. However, the effect of HTRA2 was independent of its protease activity and the interaction between HTRA2 and TRAP1 was abrogated through treatment with mitochondrial respiratory inhibitors [55]. TRAP1 overexpression is also capable of rescuing mitochondrial dysfunction-associated PINK1 and HTRA2 loss. Interestingly, HTRA2 is also a substrate of PINK1, demonstrating that further work is needed to understand the mechanistic regulation of TRAP1 by HTRA2 and the role of PINK1 in this system.

Neurofibromatosis is caused by mutation and inactivation of the Ras regulatory protein neurofibromin and is characterized by elevated Erk1/2 activity [10]. Active Erk1/2 is associated with TRAP1-SDH in the mitochondria of these cells, and Erk1/2-mediated phosphorylation of TRAP1-S511/S568 strengthens their association, suggestive of a chaperone–client relationship. Association of TRAP1 and SDH decreases SDH activity, leading to accumulation of the oncometabolite succinate [10]. TRAP1 attenuation or loss of phosphorylation at these residues prevents tumor growth, in a succinate-dependent manner [10]. Mitochondrial Erk1/2 was previously shown to antagonize PTP opening [107], perhaps indicating a role for TRAP1 phosphorylation in PTP regulation as well. Taken together, these data suggest that TRAP1 inhibition or combined TRAP1-Erk1/2 targeting may be a viable therapeutic strategy in neurofibromatosis and other cancers. 

Interaction with mitochondrially localized c-Src remains the only described TRAP1–tyrosine kinase relationship [6]. Previous work has shown that mitochondrial c-Src is involved in the phosphorylation-mediated activation of ETC Complexes I, II, and IV [108,109]. TRAP1 binds to and maintains c-Src in an inactive state, providing a potential mechanism for TRAP1 suppression of oxidative metabolism and ROS mitigation [6]. Though TRAP1 tyrosine phosphorylation is induced by c-Src expression and abrogated by c-Src inhibition, direct phosphorylation of TRAP1 by c-Src remains to be demonstrated. Taken together, TRAP1 and c-Src play opposing roles in the regulation of mitochondrial metabolism, though the reciprocal impact of c-Src on TRAP1 remains unresolved.

### 4.2. Acetylation–Deacetylation

Acetylation modulates protein–protein interactions via neutralization of Lys residues and can be reversed by the activity of deacetylases. TRAP1 directly stabilizes one such deacetylase, sirtuin-3 (SIRT3), and augments SIRT3 activity in vitro and in glioma cells [27]. Interestingly, SIRT3 overexpression was also able to rescue the effects of TRAP1 inhibition by the TRAP1 inhibitor gamitrinib [27]. One potential explanation for this observation is that SIRT3-mediated deacetylation of TRAP1 modulates TRAP1 activity or its affinity for gamitrinib, though no direct evidence was reported [27]. SIRT3 knockdown was also shown to increase ROS levels, and SIRT3 overexpression reversed an increase in ROS caused by gamitrinib [27]. Interestingly, attenuation of SIRT3 specifically destabilized TRAP1 substrates NDUFA9 (CI) and SDHB (CII), but not SIRT3 substrates SOD2 and GDH, suggesting that SIRT3-mediated deacetylation of TRAP1 is important for TRAP1 chaperone activity [27]. Interestingly, these interactions were observed in glioblastoma (GBM) cancer stem cells (CSC), which showed a preference for mitochondrial respiration over glycolysis. This work provides a new paradigm for understanding the role of SIRT3 in cancer [110]. Given this context and the known role of both proteins in regulating mitochondrial metabolism, reciprocal regulation of SIRT3 and TRAP1 may provide a positive feedback mechanism that impacts the ability of TRAP1 to chaperone its dependent proteins.

### 4.3. Nitrosylation

The PTM S-nitrosylation (SNO) is the result of the covalent addition of -NO to the thiol group of cysteine residues [111]. SNO is enzymatically catalyzed by nitrosylases and reversed by the activity of denitrosylases, including S-nitrosoglutathione reductase (GSNOR) [112]. GSNOR is commonly deleted in hepatocellular carcinoma (HCC), and GSNOR-KO mice develop HCC, linking aberrant nitrosylation to cancer [113]. TRAP1-C501-SNO was identified by mass spectrometry [54,114] and this modification was found to decrease TRAP1 ATPase activity, modulate conformational rearrangement, and promote its proteasomal degradation [54,98]. TRAP1 degradation also led to increased SDH activity, in agreement with previous work [44], and sensitized cells to SDH inhibitors, identifying TRAP1-SNO as a predictor of tumor cell response to this class of drugs [54]. It follows that mutation of this residue to TRAP1-C501S provided protection from apoptosis in the presence of nitric oxide donors, demonstrating that disruption of TRAP1-SNO is essential for its anti-apoptotic role [98]. Curiously, however, TRAP1 is overexpressed in many cancers, allowing for the possibility that TRAP1-SNO is context-specific and perhaps also under temporal regulation.

Taken together, PTMs exert influence on TRAP1 through regulating the kinetics of ATP hydrolysis and associated conformational rearrangements, interaction with client proteins, and promoting TRAP1 degradation.

## 5. Current State of TRAP1 Inhibitor Development

Inhibition of cell metabolism is a re-emerging anti-cancer strategy [115]. TRAP1 control of cellular metabolic flux and mitochondrial apoptosis outlined herein identifies TRAP1 inhibition as a potential anti-cancer therapeutic target. Efforts towards the development of ATP-competitive inhibitors for cytosolic Hsp90 have provided lead compounds for optimization to address the dual challenges of mitochondrial localization and TRAP1 specificity. Conjugation to a chemical scaffold such as the mitochondrial-targeting moiety triphenylphosphonium (TPP) is necessary to provide mitochondrial penetrance [116,117]. Specificity for TRAP1 over Hsp90 may also be a necessary consideration, as well-established Hsp90 ATP-competitive inhibitors cannot differentiate between the ATP-binding pockets, potentially leading to off-target toxicity [33].

### 5.1. Gamitrinibs

The most widely used mitochondrial Hsp90 inhibitors are gamitrinibs (G), small molecules consisting of the Hsp90 inhibitor 17-allylamino-17-demethoxygeldanamycin (17-AAG) attached to a mitochondrial-targeting moiety such as cyclic guanidinium repeats or TPP (G-G1-4 and G-TPP, respectively) [118]. These gamitrinibs have demonstrably reduced the viability of prostate [91,119,120,121,122], colon [119,123], melanoma [119,124], cervix [122,125], ovary [122], breast [118,119,121,124,125], and glioma cancers [126], particularly glioblastomas [120,124,127,128,129]. Gamitrinibs disrupt the anti-apoptotic effects of TRAP1, as evidenced by decreased mitochondrial membrane potential and increased cytochrome *c* release in G-TPP-treated PC3 prostate cancer cells [119]. Furthermore, the stability of the sensitive cytosolic Hsp90 client proteins Akt and phospho-Y416-Src was impacted by 17-AAG treatment, but unaffected by G-TPP in PC3 cells, demonstrating the selective targeting of gamitrinibs to the mitochondria [119]. A further consideration is the potential for resistance development, as PC3 cells continuously incubated with 17-AAG eventually became resistant to G-TPP, but not G-G4 [118,119]. This finding potentially suggests that the choice of mitochondrial-targeting moiety may be critically important and not necessarily limited simply to drug transport. Overall, selective TRAP1 inhibition with ATP-competitive gamitrinib derivatives remains a challenge. Further, these data emphasize the importance of understanding effectors of TRAP1 for the identification of potential combinatorial therapeutic targets to augment inhibition of TRAP1-mediated signaling pathways.

### 5.2. Purine-Scaffold Inhibitors

In addition to 17-AAG, mitochondrial targeting of the purine-scaffold Hsp90 inhibitor PU-H71 has also demonstrated efficacy against TRAP1. A TPP-conjugated derivative of PU-H71 (SMTIN-P01) showed a remarkable ability to target mitochondria over non-conjugated PU-H71 and a slight improvement in cytotoxicity over gamitrinibs [130]. Interestingly, adjustments to the length of the TPP resulted in changes in inhibitor behavior. When the TPP was modified to have a 10-length carbon chain (as opposed to the standard 6-length carbon chain), this so-called SMTIN-C10 induced structural changes to TRAP1 and demonstrated increased inhibition of TRAP1 [52]. SMTIN-C10 was found to bind to an allosteric binding site at E115 in the N-terminal domain of TRAP1, in addition to binding to the ATP pocket, resulting in TRAP1 adopting a closed formation [52]. This long linker approach was adapted for other TRAP1 inhibitors as well, including Mitoquinone. TPP-Mitoquinone has shown utility and specificity by targeting the client-binding middle domain of TRAP1 [117]. Mitoquinone has been demonstrated to have protective properties in various animal models of neurological maladies, such as traumatic brain injury [131], Huntington’s disease [132], amyotrophic lateral sclerosis (ALS) [133], and Alzheimer’s disease [134]. This finding is contradictory to the working model of TRAP1 function, especially considering that TRAP1 downregulation is observed in Alzheimer’s disease patients [135] and its overexpression is protective against oxidative stress in ALS [62]. These results highlight the need to understand the disease-specific contexts of TRAP1 function to identify appropriate disease models for the evaluation of TRAP1 inhibitors. 

### 5.3. New Inhibitors

Since their discovery, Hsp90 inhibitors have primarily targeted the ATP-binding pocket (Figure 5). This is the mechanism of the natural product geldanamycin (GA) [136,137,138] and its derivatives, as well as the first synthetic inhibitor of TRAP1, Shepherdin [139]. Shepherdin was designed by imitating the minimal Hsp90-binding sequence of Survivin (aa 79–87), an anti-apoptotic protein that binds to the N-domain of Hsp90 [140]. Consequently, Shepherdin was also found to disrupt Hsp90-ATP binding with 13 predicted sites of hydrogen bonding in the ATP pocket [139]. Modeling studies based on the structure of Shepherdin identified the small molecule 5-aminoimidazole-4-carboxamide-1-β-D-ribofuranoside (AICAR), a previously characterized AMPK activator [141,142], as a potential Hsp90 inhibitor, though its development as a scaffold for Hsp90 inhibition has not been pursued.

Though ATP-competitive Hsp90 inhibitors are still widely used, an alternative approach in hopes of achieving TRAP1 specificity over other Hsp90 family members has emerged through allosteric targeting. One example of this strategy is honokiol bis-dichloroacetate (HDCA), which is able to specifically inhibit TRAP1 by binding to an allosteric pocket within the middle domain. This pocket has a surface landscape defined by a positively charged region sandwiched between two negatively charged regions that are separated from each other by a large hydrophobic area. HDCA binds in this hydrophobic area and allosterically inhibits TRAP1 ATPase activity, but not that of Hsp90 [43].

Further, computational methods by Sanchez-Martin et al. utilized the unique asymmetry of TRAP1 to identify an allosteric pocket on the straight protomer of the TRAP1 dimer that can serve as a TRAP1-specific inhibitor binding surface [42]. Inter-domain communication is essential to the ATPase cycle of TRAP1, and previous work has shown that inhibitor-bound TRAP1 stalls in the NTD dimerized phase [143]. In agreement, the computationally identified compounds (compounds 5–7) were hypothesized to inhibit TRAP1 by reducing the ability of the ATP-binding site to communicate with the client-binding region of the middle domain. In fact, several of these small molecules were shown to decrease TRAP1 ATPase activity to a degree comparable to that of 17-AAG, while not significantly interfering with Hsp90 ATPase activity, demonstrating specificity for TRAP1 [42]. Furthermore, allosterically inhibited TRAP1 bound approximately 30% less SDHA than its control and experienced a significant increase in succinate-coenzyme-Q reductase (SQR) activity. While the tested compound did not alter cell viability, it delayed cell proliferation over a 96 h observation [42]. The successful utilization of TRAP1 asymmetry to identify unique allosteric binding pockets provides a significant starting point for future inhibitor work.

## 6. Future Perspectives

The function of TRAP1 as a regulator of cellular metabolic flux and mitochondrial apoptosis underscores a duality in which cell fate decisions are determined (Figure 6). Normal cells demonstrate basal TRAP1 expression, facilitating oxidative metabolism and programmed cell death. Dysregulation of TRAP1 expression manifests in noted hallmarks of cancer, including cell death resistance and deregulation of cellular energetics [144]. A thorough delineation of the mechanism of TRAP1 function in these roles is essential to combatting diseases of mitochondrial dysfunction, including cancer and neurodegeneration.

Though our understanding of the cellular impact of TRAP1 is coming into focus, several outstanding questions remain that are essential to our comprehension of the full scope of TRAP1 biology. (1) Is TRAP1 ATPase activity, and by extension TRAP1 chaperone function, essential for its biological activity? ATP-competitive inhibitors of TRAP1 demonstrate efficacy in cell models of cancer, suggesting that TRAP1 function is coupled to its ATPase activity; however, catalytically inactive TRAP1 mutants are able to complement TRAP1 function and revert metabolic dysfunction [26]. Reconciling these disparate observations is an ongoing challenge. (2) What is the physiological impact of TRAP1 dimeric and tetrameric forms, and is transition between these states essential for its function? Cytosolic Hsp90s are well-established dimers, and though the domain architecture of TRAP1 is similar, it remains unclear whether the TRAP1 dimer is the primary biological unit. (3) Is specific targeting of TRAP1 in cancer essential? Many existing TRAP1 inhibitors are mitochondrially targeted Hsp90 inhibitors. Though strategic inhibition of cytosolic Hsp90 has yet to demonstrate clinical success, perhaps simultaneous disruption of TRAP1 and the mitochondrial Hsp90 pool will prove efficacious [145]. (4) Can TRAP1 be used as a biomarker in cancer? Previous work has demonstrated that circulating Hsp90 can potentially be used as a biomarker in certain conditions, however the presence of circulating TRAP1 has not been evaluated [146,147,148]. Similarly, TRAP1 expression and activity is dysregulated in cancer, potentially suggesting an ability to serve as a predictive indicator of disease state. (5) TRAP1 mutations have been implicated in several conditions, including congenital anomalies of the kidney and urinary tract (CAKUT), vertebral defects, anal atresia, cardiac defects, tracheo-esophageal fistula, renal anomalies, and limb abnormalities (VACTERL), Parkinson’s disease, cardiac hypertrophy, and severe autoinflammation [55,149,150,151]. What is the structural basis for the impact of these mutations on TRAP1 function? Is mutant-TRAP1 association with these diseases a consequence of its role as a more general regulator of mitochondrial dynamics [152]? (6) Can differential PTM of TRAP1 in normal and disease states predict disease-associated phenotypes? Indeed, it has been shown that PTMs modulate TRAP1, however whether this necessarily predicts TRAP1 behavior in disease states remains to be tested. (7) Do TRAP1 PTMs compensate for a lack of dedicated co-chaperones? In the case of Hsp90, a single phosphorylation can functionally replace the loss of the yeast co-chaperone Hch1 [101]. The relevance of this mechanism for TRAP1 has not yet been investigated, however the reliance of cytosolic Hsp90 on co-chaperone interaction suggests that TRAP1 PTMs can recapitulate some co-chaperone activities. (8) Can these PTMs be specifically manipulated to alter TRAP1 function? Many cancers are associated with increased TRAP1 activity, and decreased TRAP1 activity or loss-of-function mutations contribute to the pathogenesis of some neurodegenerative diseases [153]. Previous work discussed here demonstrates that PTMs play a role in the regulation of TRAP1 stability, and TRAP1 PTMs are dysregulated in disease. High-throughput methods [154] as well as the study of cytosolic Hsp90s suggest that TRAP1 function will be regulated by a constellation of PTMs with differential incidence that correlates with disease state [3].

The literature reviewed here from several experimental systems demonstrates that in cancers that overexpress TRAP1, attenuation of TRAP1 expression or activity is sufficient to slow cell growth, and in some instances, induce apoptosis. Furthermore, nuanced studies of Hsp90 have demonstrated that PTM can modulate the efficacy of Hsp90 inhibitors [3], implying a similar framework for the application of TRAP1 inhibitors. The identification of predictive indicators of response to TRAP1 inhibition and potential targets for anti-cancer therapy in combination with TRAP1 inhibitors are two essential pieces of information that can be gained from decrypting the TRAP1 chaperone code.

## Figures and Tables

**Figure 1 biomolecules-12-00786-f001:**
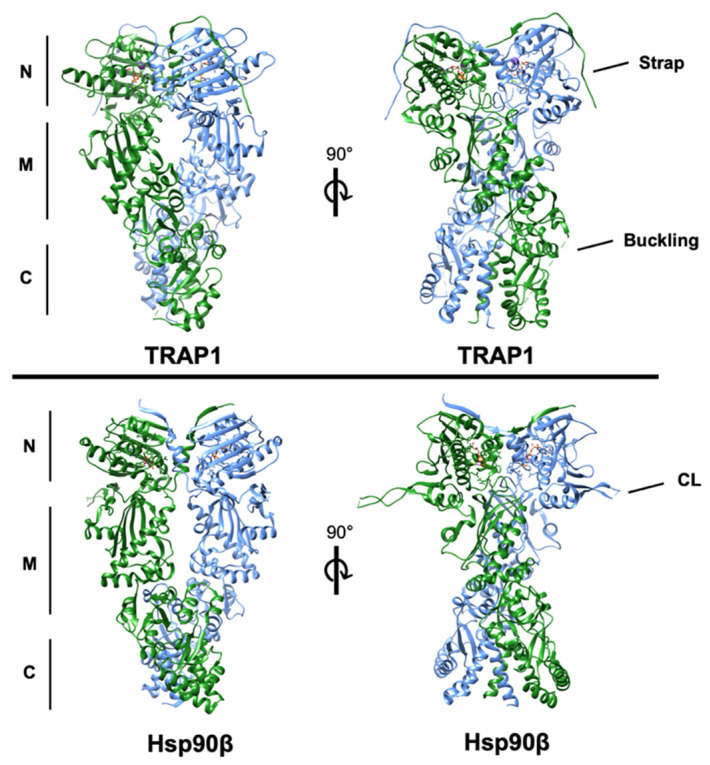
Structures of human TRAP1 (PDB: 6xg6) and human Hsp90β (PDB: 5fwp) bound to nucleotide with the conserved N-, middle-, and C-domains denoted. One protomer of each is colored blue and the second is colored green. The regulatory N-terminal extension (strap) of each TRAP1 protomer can be observed overlapping the opposite protomer. The region of TRAP1 near the M-C boundary that ‘buckles’ during conformational rearrangement is incompletely resolved in the structure. Additionally, the resolved residues of the charged linker domain (CL) of cytosolic Hsp90, which is absent in TRAP1, are labeled in the lower right quadrant.

**Figure 2 biomolecules-12-00786-f002:**
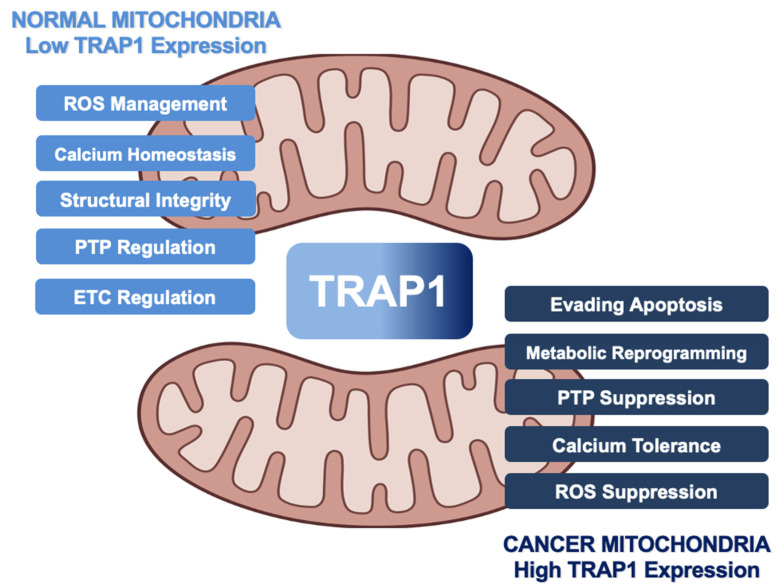
Role of human TRAP1 in mitochondria of normal cells and cancer cells. Normal expression levels (light blue) lead to TRAP1 regulation of ROS and calcium levels, integrity of cristae, function of ETC, and oversight of the PTP. As TRAP1 expression increases (dark blue), mitochondria lose calcium sensitivity, downregulate ROS, and prevent PTP opening, leading to metabolic reprogramming and evasion of apoptosis in cancer.

**Figure 3 biomolecules-12-00786-f003:**
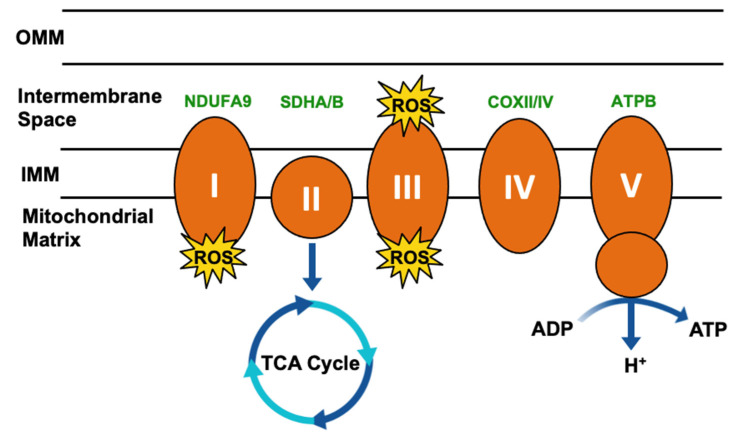
Simplified mitochondrial respiration schematic. Electron transport chain (ETC) complexes (I–V) are represented by orange ovals, and reactive oxygen species (ROS) generated as a byproduct of Complex I and III activity is represented by yellow starbursts. Succinate dehydrogenase (SDH)/Complex II connects the ETC to the tricarboxylic acid (TCA) cycle. TRAP1 interactors involved in this process have been highlighted in green.

**Figure 4 biomolecules-12-00786-f004:**
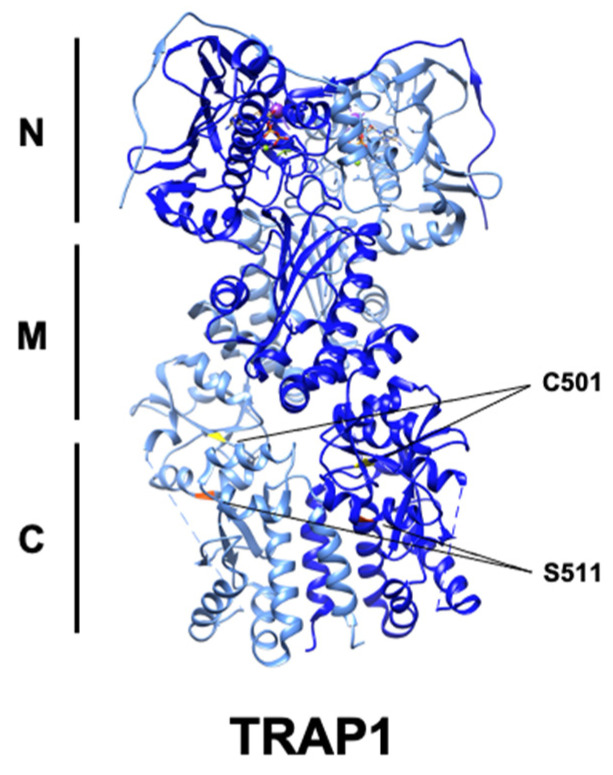
Ribbon structure of human TRAP1 (PDB: 6xg6) with known PTM sites. C501 (yellow) and S511 (red) are highlighted, while S568 is absent.

**Figure 5 biomolecules-12-00786-f005:**
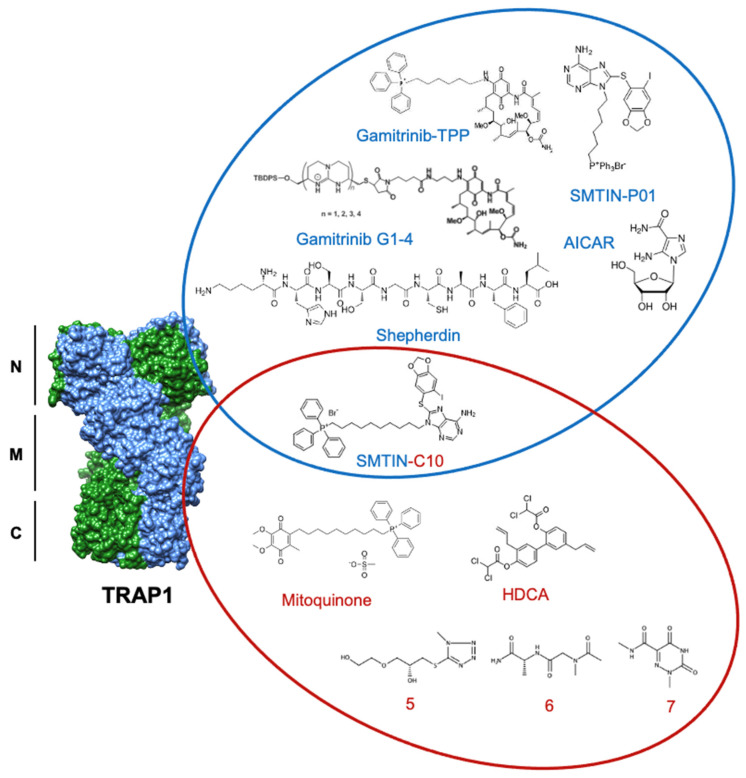
Structures of discussed TRAP1 inhibitors. ATP-competitive small molecules targeting the N-domain of TRAP1 (PDB: 6xg6) are labeled blue, while allosteric inhibitors, primarily targeting the TRAP1 middle domain, are labeled red. SMTIN-C10 is a bifunctional inhibitor, with elements of both ATP-competitive and allosteric inhibition.

**Figure 6 biomolecules-12-00786-f006:**
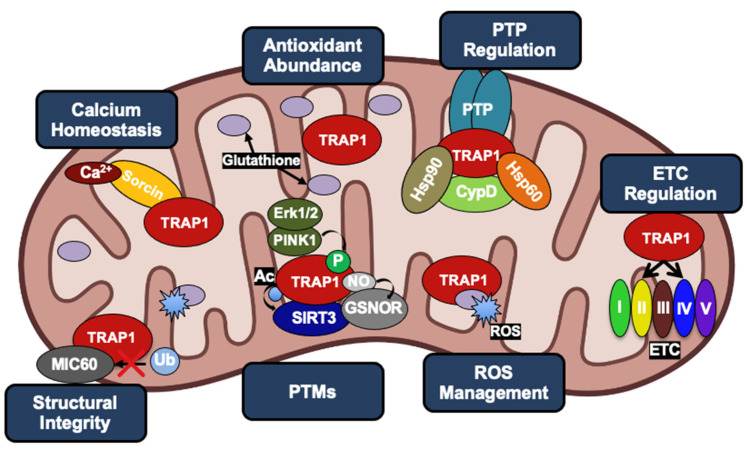
The multiple roles of TRAP1 in cancer cell mitochondria, revolving around evasion of apoptosis and metabolic reprogramming. TRAP1 acts as a chaperone for the Ca^2+^ binding protein Sorcin as well as Complexes II and IV of the ETC. Increased TRAP1 levels are associated with calcium tolerance, increased levels of the antioxidant glutathione, reduced levels of ROS, reduced levels of MIC60 ubiquitination, and in many cases, a shift towards the Warburg effect. TRAP1, along with Hsp90 and Hsp60, can form a complex with CypD to prevent opening of the PTP. TRAP1 is post-translationally modified by PINK1, Erk1/2, GSNOR, and SIRT3.

**Table 1 biomolecules-12-00786-t001:** Reported PTMs of TRAP1. Paralog identifies conserved residues in Hsp90α. GSNOR—S-nitrosoglutathione reductase, ERK—extracellular signal-regulated kinase.

Modification	Enzyme	Residue	Paralog	Impact on TRAP1	Reference
*S*-Nitrosylation	GSNOR	Cys501	Thr495	Decreased activity, proteasomal degradation	[98]
Phosphorylation	ERK1/2	Ser511	Ser505	N/A	[10]
Phosphorylation	ERK1/2	Ser568	Glu562	Increased SDH inhibition	[10]
S/T Phosphorylation	PINK1	N/A	N/A	N/A	[5]
Y Phosphorylation	Unknown, possibly c-Src	N/A	N/A	Disrupts c-Src interaction	[6]
Deacetylation	SIRT3	N/A	N/A	Increased activity	[27]

## Data Availability

Not applicable.

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
