# Peer review of "TRAP1 Chaperones the Metabolic Switch in Cancer"

_biomolecules, 2022, doi:10.3390/biom12060786_

Round 1

Reviewer 1 Report

The authors wrote an interesting and exhaustive review on the ability of TRAP1 to mediate metabolic rewiring in cancer. The manuscript is of interest for a broad audience and deserves publication.

Some typos are present in the manuscript, so the review will benefit from an accurate rereading.

Minor comments are listed below:

-Line 86. Please check this sentence: “Despite this controversy, much of the literature supports the idea that TRAP1 regulates metabolic transformation during tumorigenesis, TRAP1 overexpressed in many cancers, and TRAP1 attenuation is detrimental to tumor cell survival” 

-Lines 252-253. Please clarify what you mean by the second part of the sentence: “TRAP1 was shown to be phosphorylated by PINK1 and mediate PINK1 anti-apoptotic activity, as TRAP1 knockdown sensitized cells to loss of PINK1”. 

-Lines 257-258. The following sentence deserves some additional details to clarify the concept. HTRA2 seems to be deleterious whether the protein is present or absent. “Canonically, HTRA2 induces cell death via inhibition of IAPs (inhibitor of apoptosis proteins), and loss of HTRA2 is associated with aberrant mitochondrial function and Parkinson’s Disease (PD).” 

-Lines 402-403. Please check this sentence: “HDCA binds in this hydrophobic area and allosterically inhibitors TRAP1 ATPase activity, but not that of Hsp90 [42].”

 -Lines 452-453. The authors propose TRAP1 as a biomarker in cancer and wrote that “Previous work has demonstrated that circulating Hsp90 can potentially be used a biomarker in certain conditions”. Has TRAP1 been found in the blood of cancer patients or described as secreted by cancer cells?

Author Response

We thank the reviewer for their thoughtful evaluation of our work. At the reviewer’s suggestion, we have amended the five sections identified by the reviewer in the revised manuscript.

Reviewer 2 Report

The review titled “Chaperoning the Metabolic Switch in Cancer” by Wengert LA et al, is interesting. The role of TRAP1 and its involvement in the metabolic processes of neoplastic cells is described under various aspects. The authors describe from the biomolecular point of view the activities carried out by TRAP1 also analyzing the post-translational changes that are remarkably interesting in the metabolic dysregulation that occurs in cancer. Finally, the part relating to TRAP1 inhibitors is interesting.

In the manuscript there are several figures which, in a simple and clear way, describe and summarize what is present in the text. In this regard, I suggest the authors to create a summary figure on TRAP1 inhibitors to be included in section 5.

Finally, it is necessary to create a list, in alphabetical order, with all the numerous abbreviations present in the text.

Author Response

We thank the reviewer for careful consideration of our manuscript. We appreciate the suggested improvements and have incorporated them into the revised manuscript as Figure 5 (p.11) and ‘Abbreviations’ (p.15).

Reviewer 3 Report

The Review “Chaperoning the Metabolic Switch in Cancer” by Laura A. Wengert et al. is very interesting. 

 The authors describe the mechanisms of TRAP1 regulation in great detail, Structural basis of TRAP1 activity, Impact of TRAP1 on cancer metabolism, Post-translational regulation of TRAP1 and Future perspectives.

The title of Review Chaperoning the Metabolic Switch in Cancer”.

However, the authors throughout this review discuss recent advances in understanding the mechanisms of TRAP1 regulation, the effects of this regulation on TRAP1 function and downstream cellular processes, and the role of TRAP1 in cancer.

I recommend changing the title and resubmitting the review after changes have been made.

The minor aspects:

Point 1

Abstract - Please full expand abbreviation “TRAP1” 

Point 2

Please full expand all abbreviation in the special section.

Author Response

We appreciate the reviewer for their critical reading of our manuscript. Upon further consideration, we agree with the reviewer that the title is not representative, and have revised the title to “TRAP1 chaperones the metabolic switch in cancer.” Further, we have revised the manuscript text based on the two points made by the reviewer, including the addition of an ‘Abbreviations’ section at the end of the main text.